# Resuscitative Endovascular Balloon Occlusion of the Aorta (REBOA) in Non-Traumatic Cardiac Arrest: A Narrative Review of Known and Potential Physiological Effects

**DOI:** 10.3390/jcm11030742

**Published:** 2022-01-29

**Authors:** Carlo Alberto Mazzoli, Valentina Chiarini, Carlo Coniglio, Cristian Lupi, Marco Tartaglione, Lorenzo Gamberini, Federico Semeraro, Giovanni Gordini

**Affiliations:** Department of Anesthesia, Intensive Care and Prehospital Emergency, Ospedale Maggiore Carlo Alberto Pizzardi, 40133 Bologna, Italy; carloalberto.mazzoli@ausl.bologna.it (C.A.M.); valentina.chiarini@ausl.bologna.it (V.C.); carlo.coniglio@ausl.bologna.it (C.C.); cristian.lupi@ausl.bologna.it (C.L.); marco.tartaglione@ausl.bologna.it (M.T.); federico.semeraro@ausl.bologna.it (F.S.); g.gordini@118er.it (G.G.)

**Keywords:** REBOA, non-traumatic cardiac arrest, cardiopulmonary resuscitation, aortic occlusion, ROSC

## Abstract

Resuscitative endovascular balloon occlusion of the aorta (REBOA) is widely used in acute trauma care worldwide and has recently been proposed as an adjunct to standard treatments during cardiopulmonary resuscitation in patients with non-traumatic cardiac arrest (NTCA). Several case series have been published highlighting promising results, and further trials are starting. REBOA during CPR increases cerebral and coronary perfusion pressure by increasing the afterload of the left ventricle, thus improving the chances of ROSC and decreasing hypoperfusion to the brain. In addition, it may facilitate the termination of malignant arrhythmias by stimulating baroreceptor reflex. Aortic occlusion could mitigate the detrimental neurological effects of adrenaline, not only by increasing cerebral perfusion but also reducing the blood dilution of the drug, allowing the use of lower doses. Finally, the use of a catheter could allow more precise hemodynamic monitoring during CPR and a faster transition to ECPR. In conclusion, REBOA in NTCA is a feasible technique also in the prehospital setting, and its use deserves further studies, especially in terms of survival and good neurological outcome, particularly in resource-limited settings.

## 1. Introduction

The use of resuscitative endovascular balloon occlusion of the aorta (REBOA) for the treatment of trauma patients with active bleeding is a widely studied and accepted practice worldwide [1]. The use of REBOA for hemostatic purposes has shown promising results in terms of outcome and has recently been included in the latest guidelines for the management of patients with traumatic cardiac arrest as a rescue option in an attempt to achieve the return of spontaneous circulation (ROSC) [2]. This indication outweighs the lack of consensus on the use of REBOA in patients in extremis (no pulse) recently discussed by an expert panel [3].

In recent years, several groups have started to study the use of REBOA in the management of non-traumatic cardiac arrest (NTCA), proposing several case series and in vivo studies in animal models.

Although still widely debated [4], different centers have started to use this technique to improve the odds of ROSC with good neurological outcomes, which remain rather low.

The progressive development and implementation of extracorporeal cardiac life support (ECLS) have given rise to a series of new considerations on additional treatments for NTCA [5]; unfortunately, this technique is quite expensive [6], challenging in terms of logistics, and requires advanced skills not yet available on a large scale. Thus, it is worth considering the use of REBOA in NTCA, as this method, which is cheaper and carries a shorter learning curve, can play an important role as an adjunct treatment in settings where the implementation of an ECLS program is not currently feasible.

Moreover, the implementation of REBOA in the prehospital setting for the treatment of patients with hemorrhagic shock directly at the scene has opened further scenarios. In the above-mentioned Delphi consensus paper [3], experts could not reach an agreement on the use of aortic occlusion in out-of-hospital settings, but the discussion was limited to bleeding patients. The same year, Brede et al. demonstrated how the application of this technique can be rapidly extended to patients with out-of-hospital NTCA [7].

This paper aims to analyze the effects of aortic occlusion in patients with NTCA and to define a potential physiological rationale to support the emerging evidence in this field of study.

## 2. Materials and Methods

This paper provides a narrative review of the literature that focuses on the role of the REBOA technique in NTCA that is refractory to conventional cardiopulmonary resuscitation.

The search was conducted in Medline and Embase databases, between 1 January 1955 and 31 October 2021, using free-text terms and MeSH terms: “Cardiopulmonary Resuscitation”, “Non-Traumatic Cardiac Arrest”, “Refractory Ventricular Fibrillation”, “Coronary Perfusion”, “Hemodynamic”, plus the Boolean operator “AND” with the terms “REBOA” and “Aortic occlusion”.

The abstracts were evaluated by both authors. References for each paper found relevant were scrutinized for further articles regarding the use of REBOA in NTCA that had not emerged from the initial research. No language restrictions were adopted.

## 3. REBOA in Non-Traumatic Cardiac Arrest

Currently, the REBOA technique refers to aortic occlusion within one of three distinct zones: Zone 1 begins from the origin of the left subclavian artery and extends to the coeliac artery; Zone 2 includes the portion from the coeliac artery to the most distal renal arteries; and Zone 3 extends from the distal renal artery to the aortic bifurcation.

While Zone 3 is usually used to manage pelvic and lower-extremity hemorrhage, Zone 1 is measured to the xiphoid and is used in torso hemorrhage. Zone 2 has no current indication.

The hypothesis of the use of Zone 1 REBOA in NTCA has been put forward in recent years [8], proposing as a rationale the increased perfusion of the organs upstream of the occlusion during CPR.

The first feasibility trial on the use of REBOA in out-of-hospital cardiac arrest (OHCA) was published by Brede et al. [7] in 2019; the study reported 60% ROSC, versus the 14% from the international literature. Two further important data were the increase in etCO_2_ within 60 s of balloon insufflation, and that the procedure was successful in 100% of cases without affecting the quality of advanced cardiac life support (ACLS).

Several case reports were subsequently reported, and the following year, Levis et al. [9] published a pilot trial with 15 patients reporting only two ROSCs but no increase in etCO_2_. In this study, a significant increase in cerebral oxygenation was observed.

Our study group recently published a prospective cohort study [10] on 20 patients with traumatic and non-traumatic cardiac arrest in whom REBOA was used as an adjunctive treatment. The results confirmed the increase in etCO_2_ after balloon inflation, and a potential benefit in terms of odds of ROSC in a group of patients not eligible for ECLS.

Recently, a nice review by Nowadly et al. [11] gathered all the evidence regarding the use of REBOA as a new technique in NTCA, also exploring the salient physiological aspects. Several key points emerged from this work; we will review these aspects below, adding new possible effects that aortic occlusion can have to promote ROSC. The potential physiological effects and logistical advantages due to this technique are represented schematically in Figure 1.

## 4. Physiological Effects of REBOA in Non-Traumatic Cardiac Arrest

### Coronary Perfusion

Myocardial reperfusion is necessary for the heart to restart in both non-shockable and shockable rhythms [12,13], and thus, efforts to optimize cardiopulmonary resuscitation [14] are aimed to increase coronary perfusion pressure (adequate depth and rate of chest compressions, minimizing interruptions), including the introduction of mechanical CPR.

Coronary perfusion pressure (CPP) is defined as the difference between right atrial pressure (RAP) and mean arterial pressure (MAP) measured in the aortic bulb.
CPP = MAP − RAP

Old studies [15] have suggested that a CPP value of at least 15–25 mmHg is needed to achieve ROSC. However, these studies led to the measurement of CPP after prolonged periods of cardiac arrest. More recent studies [16] on animal models have suggested that CPP should be considered not as a threshold value to be reached but as a “total dose” in relation to the duration of resuscitation. This leads one to think of a defined amount of blood with which to perfuse the myocardium during resuscitation that reasonably depends on anthropometric parameters and inter-individual biological variability.

Considering that, in cardiac arrest conditions, RAP increases [17] due to stasis and chest compressions, it is necessary to increase MAP to achieve adequate CPP.

The hemodynamic changes induced by aortic occlusion have been described by analyzing the backward waves [18] caused in the aortic flow, and they are potentially useful for perfusion of the supra-aortic trunks [19]; a recent study also demonstrated that aortic occlusion in Zone 3 (distal to renal arteries) has little impact on those waves. This is the reason why balloon placement in Zone 1 is recommended in NTCA.

The use of Zone 1 REBOA reduces the distribution of the systolic stroke volume generated during the chest compressions, increasing MAP [20]. In addition, animal studies [21] have shown that the increase in MAP is persistent and immediate after inflation, unlike adrenaline, which takes about 60 s after administration to obtain a similar effect.

In a porcine model of NTCA, a significant increase in diastolic blood pressure and thus coronary perfusion was observed during CPR [22].

Finally, it is plausible that aortic occlusion also improves coronary perfusion during the relaxation phases of cardiac massage or backward passes by increasing the Windkessel effect [23] due to the aortic wall’s elastic recoil.

## 5. Potential Effects of REBOA in Non-Traumatic Cardiac Arrest

### 5.1. Baroreceptor Reflex

The baroreceptor reflex is a feedback loop homeostatic mechanism aimed at maintaining the stability of blood pressure and, more generally, of cardiac output. The *primum movens* resides in the baroreceptors located in the wall of the aortic arch and in the carotid sinuses, which respond to distension due to an increase in systolic pressure. The signal is then transmitted to the brainstem, causing inactivation of the sympathetic branch and activation of the parasympathetic branch of the autonomic nervous system [24]. The whole reflex can rely on several feedback loops for continuous balancing between the two branches.

During high-quality CPR, a sudden aortic occlusion could cause distension of the aortic and carotid walls, leading to a stimulation of the parasympathetic branch and, indeed, of the vagus nerve, as observed in animal heart-beating models [25]. This finding is further supported by the evidence of an increase in aortic and carotid pressure after REBOA inflation in animal models.

This should be of some importance, as vagal stimulation effects on ventricular arrhythmias have been widely studied and have shown to increase the threshold of onset of ventricular fibrillation. In animal models, it has been observed that the induction of ventricular fibrillation is more difficult during vagus nerve stimulation [26,27].

There are currently no data on vagal activity during CPR, and it would, in any event, be difficult to estimate the effectiveness of this reflex in conditions of severe hypoperfusion of the brainstem; however, in cases where there are clinical signs of perfusion of the medulla, where the regulatory centers are located, such as gasping [28], one can hypothesize a residual functioning.

It is rather unlikely that it has a role in the termination of ventricular fibrillation, considering that, to our knowledge, this has never been observed.

It is possible that the triggering of a baroreflex due to aortic occlusion may contribute, as part of all resuscitation maneuvers, to predisposing myocardial tissue to a recovery of contractile function in the event of a persistent malignant arrhythmia.

### 5.2. Adrenaline Blood Concentration

Studies in animal models have shown that aortic occlusion drastically reduces the return of blood from the inferior vena cava [19], which is a logical consequence of the interruption of arterial blood flow in the subdiaphragmatic aorta, causing de facto complete visceral ischemia. As a result of this major change, the circulating blood pool is reduced by a large proportion and limited to the districts supplied by the supra-aortic trunks [29,30,31].

Normally, adrenaline boluses are administered via venous lines positioned in the patient’s limbs; placing them in the arms seems mandatory to make sure the drugs are drained into the superior vena cava in the case of aortic occlusion in Zone 1.

Moreover, it is reasonable to think that the dose of adrenaline administered enters the circulation in a momentarily reduced quantity of blood flow, as explained above, making, at least initially, plasma concentration higher than in a conventional situation, and thus decreasing the volume of drug distribution and potentially increasing its efficacy.

In contrast, there is a marked reduction, if not a complete interruption, in adrenal gland perfusion, which has been observed to secrete copious amounts of catecholamines during cardiac arrest in response to low flow.

Given all these aspects and the known detrimental effects that β-adrenergic stimulation can have on myocardial function and cerebral perfusion [21,32,33], it is plausible to aim for a reduction in adrenaline dosages once aortic occlusion has been achieved, and further studies should better define this aspect.

Lower circulating adrenaline blood levels may result in greater hemodynamic stability and lower arrhythmogenic risk after ROSC.

## 6. Pitfalls and Caveats

While it is far from the purpose of this article to describe practical and technical aspects concerning REBOA positioning, it is nevertheless imperative to consider the risk of complications when describing the physiological aspects of an invasive maneuver.

The most important to mention are those related to ischemia [34,35], mainly of the ipsilateral limb of the cannulated femoral artery, and vascular [36] damage of the femoral artery or aorta.

These risks are mitigated by the use of smaller catheters and consequently smaller sheaths [37,38], guidewire-free catheters, and atraumatic tips.

Regarding the ischemia below the balloon, in the context of hemorrhage, noticeable results have been achieved by partial occlusion (p-REBOA) [39,40], which limits the bleeding allowing a minimum but sufficient perfusion of the splanchnic organs [41]. This technique is not practicable during NTCA but could be useful when ROSC is achieved. There is indeed a high risk of reperfusion-ischemia syndrome after ROSC and deflation of the balloon [36], so the use of a p-REBOA approach with gradual deflation and vasopressor support may be considered, even in this case.

## 7. Discussions

The use of the REBOA technique in NTCA could be a potential future change to what we now consider the standard management of CPR.

The potentially positive effects are twofold: firstly, the physiological changes that aortic occlusion causes during CPR; secondly, the additional features that a catheter itself can bring in addition to standard treatment and monitoring.

Improved coronary perfusion due to REBOA deployment is already well documented. The same kind of improvement has been demonstrated for cerebral perfusion. Increasing perfusion of the heart and brain is the main goal of CPR [42], and it may have a positive impact on the rate of ROSC.

Moreover, aortic occlusion could counteract the negative effects that adrenaline administration has been shown to have on cerebral perfusion during CPR by increasing perfusion pressure in the carotid arteries and thus the intracranial circulation.

Additionally, taking into account the possible effect on the concentration of the drug itself, one could also hypothesize a possible reduction in dosages following aortic occlusion.

On the other hand, we should consider that, once a catheter has been positioned, if an arterial line is available on the device, it is possible to inject drugs and establish infusions of fluids, even cold ones. It would be possible to take blood samples too.

Some drugs that are ineffective, such as nitrates [43], may show efficacy if administered directly into the aorta and with a reduction in systemic effects due to occlusion.

Of paramount importance is that this type of catheter provides the possibility to measure arterial pressure in the aortic arch in real-time, giving even more precise data on the perfusion of the heart and brain. A pressure wave can help to distinguish a true from a false PEA, and, combined with the etCO_2_ value, suggest the presence of a possible pulmonary embolism.

If a central venous catheter is inserted or, better still, a pulmonary artery catheter, we can estimate CPP, thus adding a potential parameter to improve the quality of CPR.

Additionally, it is worth mentioning that a catheter sheath can be used for cannulation and thus enable a faster transition to ECMO.

Furthermore, one has to consider the possible complications which, as mentioned above, can potentially be quite serious. It should be borne in mind that if this maneuver is performed by skilled professionals, it will lead to fewer complications [36], and the continuous improvement of equipment can play an important role in reducing the rate of vascular damage. According to the authors’ experience, it is essential that the REBOA technique is initially implemented in the intra-hospital setting and, at a later stage, in the pre-hospital setting involving a team of professionals working in both settings in order to ensure homogeneous and consistent management of patients undergoing aortic occlusion. Maintenance training of all staff prevents skill deterioration.

Finally, another aspect that should be addressed in future studies is the optimal timing of positioning, since earlier positioning could increase the likelihood of ROSC for the supra-mentioned mechanisms, but, on the other hand, potential benefits should be weighed against the known risks of this technique.

## 8. Conclusions

The use of REBOA during CPR in patients with NTCA is a topic that is gaining increased interest worldwide. Several studies have already been performed, and a new large RCT is underway [44].

The feasibility and potential cost-effectiveness of this addition to the established resuscitation protocols call for further studies with larger populations. Meanwhile, reliable methods and scores [45] for training professionals in emergency catheter placement are being studied; the ease with which this technique can also be used in the pre-hospital setting is one of its greatest strengths.

As mentioned above, ECLS already plays a key role, but it is not currently applicable on a large scale. It is not the intention of the authors to consider REBOA as a substitute for ECMO in patients with NTCA, but we believe it deserves further study as an addition to improve outcomes for an increasing number of patients, and also in resource-limited settings.

## Figures and Tables

**Figure 1 jcm-11-00742-f001:**
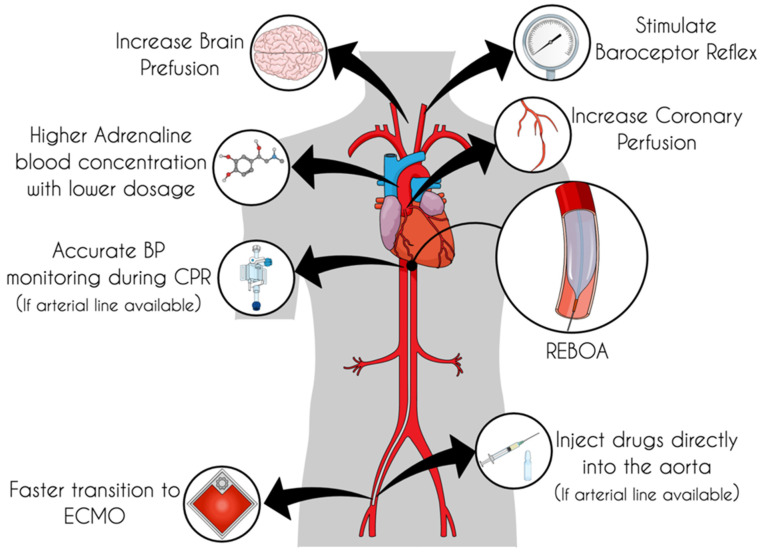
Potential physiological effects of REBOA during NTCA and the advantages of having the catheter placed into the aorta. REBOA: Resuscitative Endovascular Balloon Occlusion of the Aorta, NTCA: Non-Traumatic Cardiac Arrest, ECMO: Extracorporeal Membrane Oxygenation.

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
