# Peer review of "Resuscitative Endovascular Balloon Occlusion of the Aorta (REBOA) in Non-Traumatic Cardiac Arrest: A Narrative Review of Known and Potential Physiological Effects"

_jcm, 2022, doi:10.3390/jcm11030742_

Round 1

Reviewer 1 Report

My questions and comments were adequatly adressed in this revision, I don't have further remarks. I would like to concrate the authors for their work!

Author Response

The authors would like to thank the reviewer for the valuable suggestions.

Reviewer 2 Report

This review is a well written manuscript which adds something to the existing literature as the REBOA technique potentially improves care for NTCA patients. 

Please change title to narrative review. It is not clear in the title what exactly this manuscript is aiming to do. 

Search strategy a little weak. Please ask an expert to improve. Did the authors look within references of papers they found? 

Can the authors add something to the manuscript about positioning the REBOA in a prehospital setting and in a hospital setting? X ray needed? How important to really positioning in Zone 1 regarding the working mechanism in cardiac arrest (figure 1)

Please reference: Borger van der Burg 2019 Delphi paper in Injury. Please elaborate on this paper. 

Minor: in discussion, can the authors add something on patients with a traumatic cardia arrest? Is the REBOA not indicated in this group? 

Author Response

(The authors gave the same response as above.)
